# KdpD is a tandem serine histidine kinase that controls K$^+$ pump KdpFABC transcriptionally and post-translationally

Jakob M. Silberberg[1], Sophie Ketter[1], Paul J. N. Böhm [1], Kristin Jordan[1], Marcel Wittenberg [1], Julia Grass[1] & Inga Hänelt [1] ✉

Two-component systems, consisting of a histidine kinase and a response regulator, serve signal transduction in bacteria, often regulating transcription in response to environmental stimuli. Here, we identify a tandem serine histidine kinase function for KdpD, previously described as a histidine kinase of the KdpDE two-component system, which controls production of the potassium pump KdpFABC. We show that KdpD additionally mediates an inhibitory serine phosphorylation of KdpFABC at high potassium levels, using not its C-terminal histidine kinase domain but an N-terminal atypical serine kinase domain. Sequence analysis of KdpDs from different species highlights that some KdpDs are much shorter than others. We show that, while *Escherichia coli* KdpD's atypical serine kinase domain responds directly to potassium levels, a shorter version from *Deinococcus geothermalis* is controlled by second messenger cyclic di-AMP. Our findings add to the growing functional diversity of sensor kinases while simultaneously expanding the framework for regulatory mechanisms in bacterial potassium homeostasis.

K$^+$ homeostasis is central to enabling microbial survival under changing conditions. In many prokaryotes, high-affinity K$^+$ uptake at low extracellular K$^+$ concentrations is facilitated by the active K$^+$ pump KdpFABC[1–3]. Its chimeric composition of a K$^+$ channel and a P-type ATPase allows the complex to accumulate K$^+$ with an apparent affinity of 2 μM[4]. KdpFABC is particularly required under extreme K$^+$ limitation, while at increased extracellular concentrations toxic amounts of K$^+$ would be accumulated by its constant pumping[2]. Therefore, the expression of *kdpFABC* in *Escherichia coli* is controlled by the two-component system (TCS) KdpDE, which comprises the sensor histidine kinase (HK) KdpD and the transcription factor KdpE[5] (Fig. 1). KdpD senses extra- and intracellular K$^+$ levels[6,7]. At low K$^+$ concentrations, auto-phosphorylated KdpD phosphorylates KdpE, which, upon dimerization, activates the transcription of *kdpFABC*[8,9]. When viable K$^+$ conditions at which other transport systems can mediate K$^+$ uptake are restored, KdpFABC is no longer needed. At these conditions, KdpD switches to a phosphatase activity, dephosphorylating KdpE to stop continued transcription of *kdpFABC*[10,11].

Notably, the transcriptional control by KdpDE does not address KdpFABC complexes already in the membrane. If left unregulated, these would continue to excessively accumulate K$^{+2}$. To prevent this, KdpFABC is inhibited at K$^+$ concentrations above 2 mM via a phosphorylation of residue S162 of the P-type ATPase KdpB[12–15] (All residue numbers refer to *E. coli* KdpD and KdpFABC unless otherwise specified). This process appears to be irreversible, and prevents KdpFABC from progressing through its normal catalytic cycle[14,15]. Full inhibition was observed in situ two minutes after exposure to K$^{+12}$. However, the source of this phosphorylation is yet to be determined.

While phosphorylation is best characterized in eukaryotes, bacterial protein kinases have increasingly come into focus. They can be separated into five categories: eukaryotic-like serine/threonine kinases (eSTKs or Hanks kinases), arginine kinases, HKs, bacterial tyrosine

---

[1]Institute of Biochemistry, Biocenter, Goethe University Frankfurt, Max-von-Laue-Straße 9, 60438 Frankfurt/Main, Germany.
✉e-mail: haenelt@biochem.uni-frankfurt.de

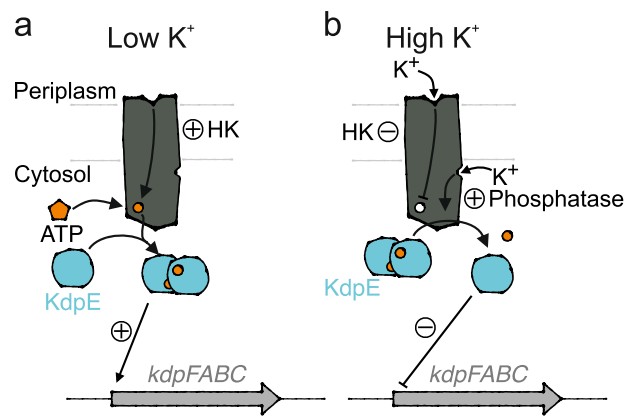

**Fig. 1 | Regulation of kdpFABC transcription by KdpDE in E. coli.** $K^+$-dependent expression of the *kdp* operon is controlled by histidine kinase KdpD and transcription factor KdpE. **a** At low $K^+$ levels, KdpD's histidine kinase activity leads to its autophosphorylation, which subsequently mediates the phosphorylation and dimerization of KdpE that activates *kdpFABC* transcription. **b** At high $K^+$ levels, $K^+$ binding to KdpD's extracellular sensor inhibits the histidine kinase. $K^+$ binding to the intracellular sensor stimulates KdpD's phosphatase activity, dephosphorylating KdpE and ending the transcription of the *kdp* operon.

kinases (BYKs), and atypical serine kinases (ASKs)[16]. eSTKs have the same characteristic motifs described for eukaryotic serine/threonine kinases and have been found to control metabolism in a wide range of bacterial species, but not in *E. coli*[17–19]. Arginine kinases have been implicated in control of bacterial protein degradation[20]. HKs are sensors that autophosphorylate at a histidine, from where the phosphate is transferred to an aspartate on a response regulator to regulate gene expression[21]. BYKs catalyze the phosphorylation of tyrosines via three Walker motifs, which are canonically involved in ATPase activities but here function as kinase domains[22]. The same active site, albeit only containing a Walker A and a Walker B motif, is found in ASKs[16]. Members of this family regulate diverse metabolic processes by serine phosphorylation[23,24]. Notably, a large number of known bacterial phosphorylation targets still lack an identified kinase[25,26].

Previously proposed mechanisms for KdpB$_{S162}$ phosphorylation include autocatalysis by KdpFABC or an unknown serine kinase that could be activated, for instance, by HK KdpD, whose low/high $K^+$ switch corresponds exactly to the concentration range at which KdpFABC inhibition occurs[12,14]. Here, we show that KdpD in fact itself acts as a tandem serine histidine kinase that both controls the transcription of *kdpFABC* and directly phosphorylates KdpB for post-translational inhibition. By analyzing the inhibition of KdpFABC in the presence of different KdpD constructs in vivo and in vitro, we show that the Walker A/B motif of the N-terminal KdpD domain functions as an ASK that phosphorylates KdpB$_{S162}$ and thus inactivates KdpFABC at high $\kappa^+$ conditions. By further analyzing the diversity of the domain organization of KdpD across prokaryotic species, we found that the ASK is actually more conserved than the HK and therefore may be the most determinant function of KdpD, and that the ASK activity of KdpDs from different species is controlled either directly by the $K^+$ concentration or by the binding of the second messenger c-di-AMP.

## Results

### KdpD mediates the phosphorylation of KdpB$_{S162}$

The inhibition of KdpFABC via KdpB$_{S162}$ phosphorylation at high external $K^+$ concentrations is essential to prevent $K^+$ toxicity[2,14]. Here, we set out to identify players responsible for the regulatory phosphorylation of KdpFABC. To characterize the inhibition level of KdpFABC, we tested the rate of ATP hydrolysis in an ATPase assay and the phosphorylation level of KdpB using a phosphoprotein gel stain (Fig. 2, Supplementary Fig. 1). KdpFABC was produced at high $K^+$

conditions in *E. coli* LB2003 cells, which lack *kdpFABC* but still express *kdpDE*, or in *E. coli* TK2281 cells, in which the entire *kdpFABCDE* operon is deleted. KdpFABC purified from *E. coli* LB2003 cells showed a low residual ATPase activity and a high level of KdpB phosphorylation, as previously reported[27,28]. When the inhibitory phosphorylation was prevented by mutation KdpB$_{S162A}$[13], ATP hydrolysis was increased six-fold, and no phosphorylation was observed. To rule out an auto-catalyzed phosphorylation of KdpB$_{S162}$, ATP binding to KdpFABC was prevented by point mutations KdpB$_{F377A/K395A}$[29]. The resulting construct showed no ATPase activity but retained the phosphorylation of KdpB, indicating that KdpB$_{S162}$ phosphorylation is mediated by a separate kinase.

It was hypothesized that KdpD, which regulates the transcription of *kdpFABC*, could also control a kinase that phosphorylates KdpB$_{S162}$. To test this hypothesis, we purified KdpFABC from the *ΔkdpDE E. coli* strain TK2281 grown at high $K^+$. KdpFABC produced under these conditions showed a high ATPase activity and no phosphorylation of KdpB, indicating that the inhibition was lifted (Fig. 2). Thus, KdpDE must be involved in KdpFABC inhibition. We considered three possible mechanisms for the regulation: through a separate kinase activated by KdpD, through a direct phosphorylation by KdpD, or through a separate kinase activated by KdpE. To distinguish the latter option from the first two, KdpD was individually reintroduced in *E. coli* TK2281 cells via expression from an orthogonal plasmid. Successful production of KdpD constructs was verified by in-gel fluorescence of whole-cell samples (Supplementary Fig. 2). KdpFABC purified from these cells was again inhibited, with an even lower ATPase activity than samples purified from *E. coli* LB2003 cells, and, coherently, showed a high phosphorylation level of KdpB. The stronger inhibition likely stems from the over-expression of KdpD compared to the low expression under the native promoter in *E. coli* LB2003 cells[12,14,30]. Thus, KdpB$_{S162}$ phosphorylation appears to be solely dependent on KdpD, and does not require the response regulator KdpE.

To elucidate the mechanism by which the phosphorylation is carried out, directly or indirectly, we next investigated the role of the individual domains of KdpD, focusing on those that potentially could mediate a phospho-transfer directly. The basic requirements for a phospho-transfer are proximity of the nucleotide or phospho-intermediate and the side chain that needs to be phosphorylated, their polarization, and the exclusion of water. KdpD consists of a four-helix transmembrane (TM) domain with two large cytosolic segments N- and C-terminally (Fig. 2 b). The N terminus comprises the KdpD domain followed by a Usp (universal stress protein) domain, while the C terminus harbors a GAF (cGMP-specific phosphodiesterases, adenylyl cyclases and FhlA) domain followed by a transmitter domain. Of these domains, the KdpD domain and the transmitter domain contain ATP binding sites that could serve the phospho-transfer to KdpB. The transmitter domain comprises a dimerization and histidine phospho-transfer (DHp) domain, and a catalytic and ATP-binding (CA) domain, and is conserved among HKs[21]. It forms the catalytic core of the HK: the CA domain binds ATP, positioning it for autophosphorylation of a histidine in the DHp domain. In *E. coli* KdpD, H673 is phosphorylated at low $K^+$. Subsequently, the phospho-histidine intermediate transfers its phosphate to KdpE[11]. The KdpD domain on the other hand is highly conserved among KdpDs from different species but not found in any other sensor kinase[31]. ATP binding to the Walker A-like motif in the KdpD domain has been implicated in regulating the phosphatase activity of KdpD at high $K^+$[32]. This motif diverges from the classical Walker A sequence by one extra residue (G-X$_5$-GKT instead of G-X$_4$-GKT)[33]. Additionally, the KdpD domain also contains a Walker B motif[31].

To test whether the phospho-histidine intermediate is used to phosphorylate KdpB$_{S162}$, the catalytic phospho-histidine was precluded by the mutation KdpD$_{H673A}$ (Fig. 2). KdpFABC produced in this background retained the inhibition and phosphorylation, indicating that the phospho-histidine is not involved in KdpB$_{S162}$ phosphorylation.

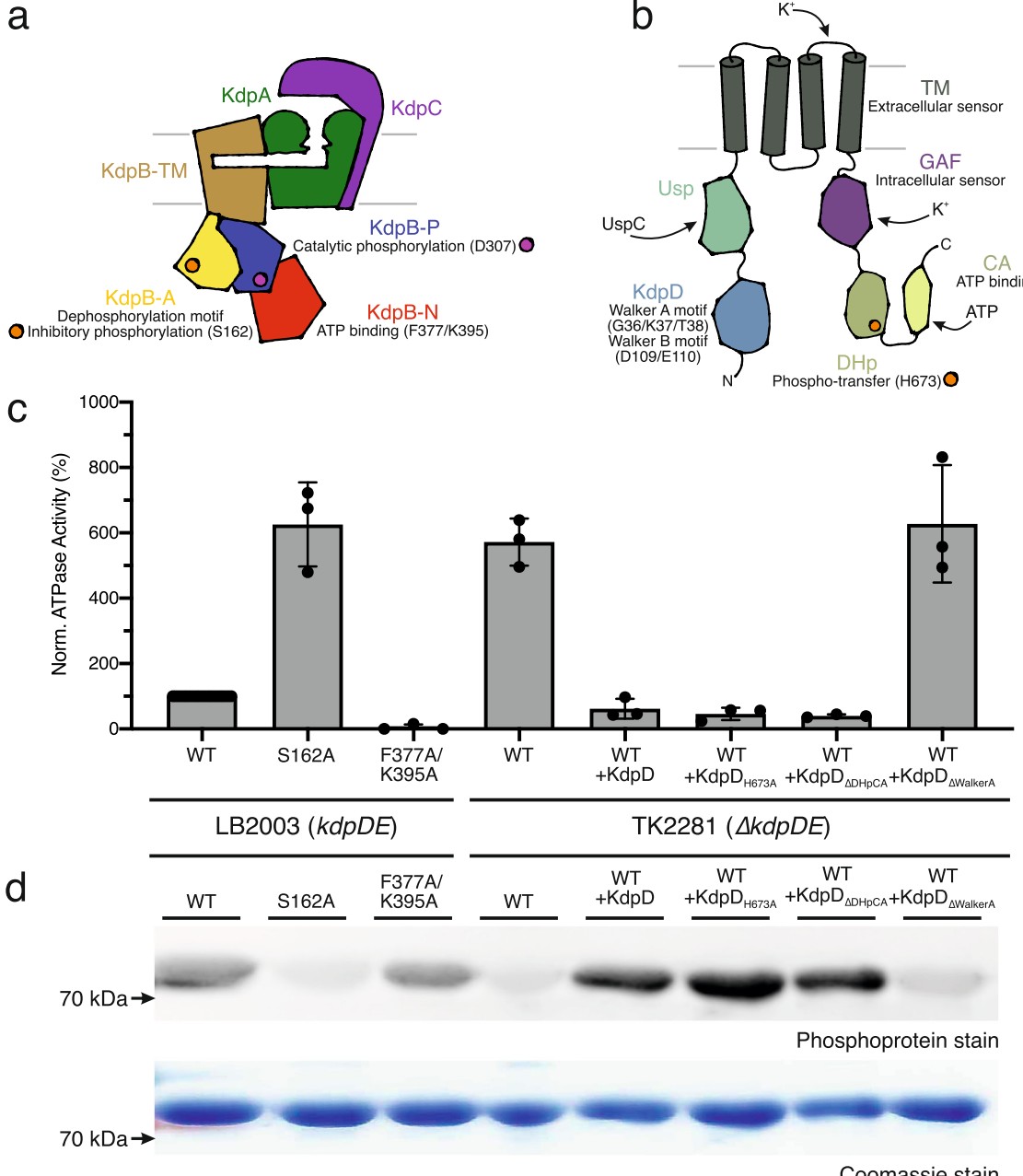

**Fig. 2 | KdpB_{S162} phosphorylation by KdpD involves the N-terminal KdpD domain. a** Schematic architecture of the KdpFABC heterotetramer. **b** Schematic architecture of a KdpD monomer. **c**, **d** Inhibition of KdpFABC by KdpD. KdpFABC constructs were purified from *E. coli* LB2003 cells, which natively express *kdpDE*, or from *E. coli* TK2281 cells, in which the *kdpFABCDE* operon is deleted. In the latter, KdpD variants were reintroduced as indicated by expression from an orthogonal plasmid. Normalized ATPase activity (**c**) of purified KdpFABC constructs relative to the activity of WT KdpFABC extracted from *E. coli* LB2003 cells, and phosphorylation state (**d**) of KdpB visualized by phosphoprotein gel stain. Bars in (**c**) indicate the mean and error bars denote the standard deviation from biological triplicates (*n* = 3). The Coomassie stain indicates comparable amounts of protein loaded per lane in the phosphoprotein stain in (**d**). Full phosphoprotein and Coomassie stains are shown in Supplementary Fig. 1, successful co-expression of KdpD variants is shown in Supplementary Fig. 2.

Alternatively, the two ATP-binding sites, the HK catalytic site in the CA domain or the Walker motifs in the N-terminal KdpD domain, could directly transfer the phosphate from ATP to KdpB. To test the role of these two sites, ATP binding was individually abolished by deleting the catalytic DHp and CA domains (KdpD_{Δ663-894}), referred to as KdpD_{ΔDHpCA}, or by introducing the mutations KdpD_{G36A/K37A/T38C} in the Walker A motif, referred to as KdpD_{ΔWalkerA}[32]. The deletion of the catalytic DHp and CA domains showed no effect on KdpFABC inhibition; ATPase activity remained low and KdpB showed high phosphorylation (Fig. 2). Mutating the Walker A motif in the KdpD domain, on the other hand, abolished the inhibition of KdpFABC and the

phosphorylation of KdpB_{S162}. Thus, the Walker motifs in the KdpD domain appear to be essential for the inhibitory phosphorylation.

## KdpB_{S162} phosphorylation is mediated by KdpD's N-terminal domain in an atypical serine kinase mechanism

While the co-expression studies in vivo indicate an important role of KdpD in KdpFABC inhibition, it is not possible to exclude a separate intermediary between the sensor kinase and KdpB. To show that the phosphorylation occurs directly between the KdpD domain and KdpB, KdpFABC and KdpD variants were purified separately from *E. coli* TK2281 cells (Supplementary Fig. 3) and mixed in vitro in the presence

of 400 mM KCl and 5 mM ATP to stimulate the serine kinase activity. All KdpFABC constructs featured the mutation $KdpB_{D307N}$ to preclude the catalytic phosphorylation of the P domain. KdpB phosphorylation was analyzed and quantified by phosphoprotein gel stain (Fig. 3 a, Supplementary Fig. 4). Without KdpD, no phosphorylation was observable. Upon incubation with full-length KdpD, the phosphorylation was fully restored. $KdpFAB_{S162A/D307N}C$ mixed with full-length KdpD showed no phosphorylation for KdpB, indicating that the signal was specific for $KdpB_{S162}$ phosphorylation (Supplementary Fig. 4). Thus, phospho-transfer must occur directly from KdpD to KdpB, making KdpD a tandem serine histidine kinase.

To show that the N-terminal KdpD domain itself catalyzes the phosphorylation, a truncated construct with only this domain ($KdpD_{1-230}$, denoted as $KdpD_{NTD}$) was purified and incubated with KdpFABC. This truncated construct remained able to phosphorylate KdpB, indicating that the domain acts as a functional module within the sensor kinase that can by itself carry out a kinase activity. In fact,

phosphorylation by $KdpD_{NTD}$ was stronger than by full-length KdpD when applied at equimolar concentrations, which could suggest that the other domains of KdpD modulate the serine kinase activity. Alternatively, it is also possible that the soluble domain more easily interacts with KdpB in vitro, while full-length KdpD is hampered by a clash of the two detergent micelles.

From sequence analysis and an AlphaFold structural prediction of the complex between $KdpD_{NTD}$ and the KdpB A domain[34,35], three conserved common ATP-binding motifs, characteristic for an ASK, are apparent: Walker A and Walker B motifs for coordination of the β- and γ-phosphates, and a tryptophan/arginine pair ($KdpD_{W39/R190}$) for the coordination of the adenine (Fig. 3b, c; Supplementary Fig. 5). Mutating the Walker A motif or the adenine coordination motif ($KdpD_{\Delta WalkerA}$ or $KdpD_{W39A/R190A}$, respectively) almost completely abolished phosphorylation of KdpB (Fig. 3), likely by preventing ATP binding. Whereas the Walker A motif is essential for ATP binding, the Walker B motif is responsible for catalysis of ATP hydrolysis in Walker

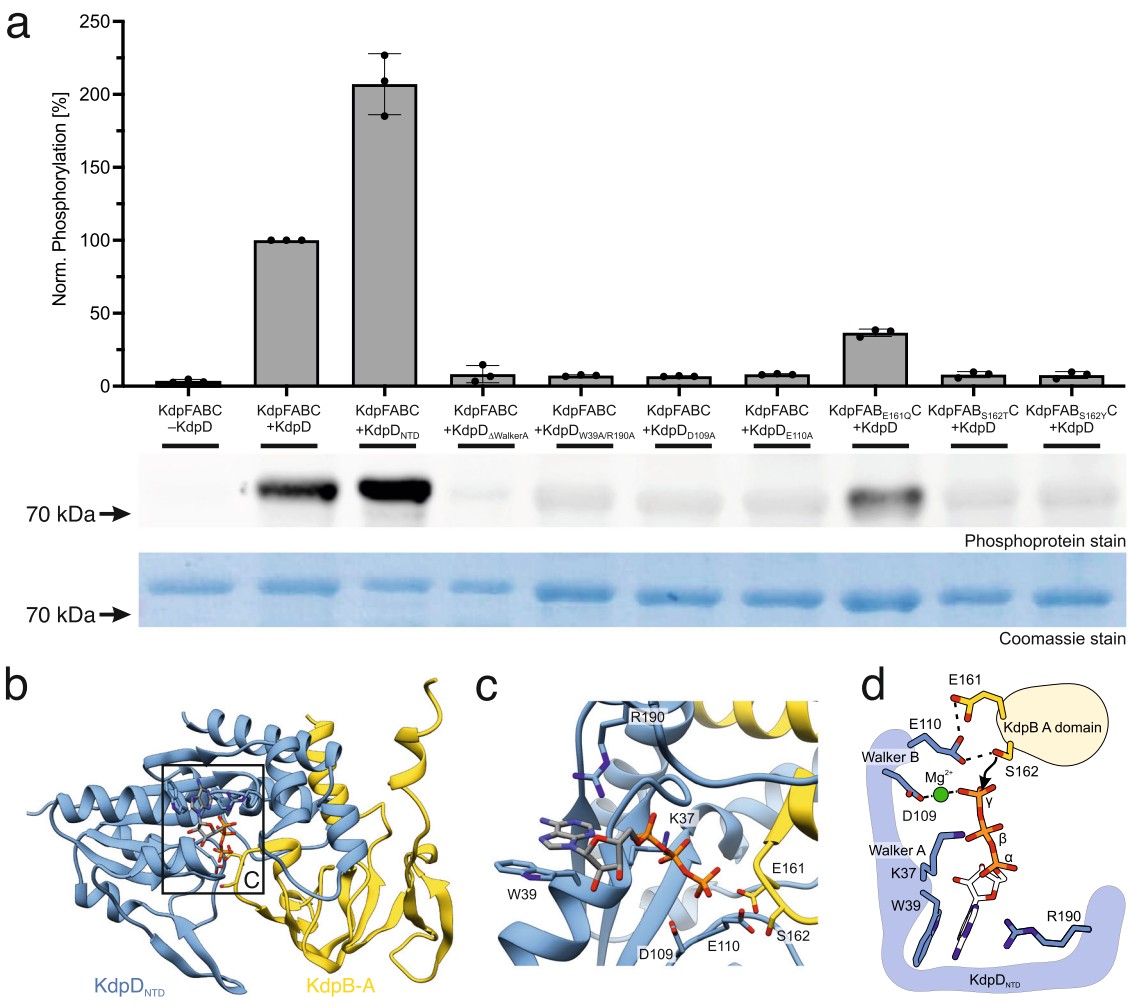

**Fig. 3 | ASK-type phosphorylation of KdpB by KdpD.** KdpFABC and KdpD constructs were purified separately from *E. coli* TK2281 cells (see Supplementary Fig. 3) and mixed in vitro in the presence of 400 mM KCl and 5 mM ATP. KdpB phosphorylation levels after 30 min were analyzed by SDS-PAGE and subsequent phosphoprotein gel stain. All KdpFABC constructs featured the mutation $KdpB_{D307N}$ to preclude the catalytic phosphorylation of the P domain. **a** Phosphoprotein gel stain of purified KdpFABC separated by SDS-PAGE, indicating the phosphorylation state of KdpB. The corresponding Coomassie stain shows the total protein amount loaded per lane. Phosphoprotein stains were quantified with ImageJ and normalized to the quantification of the Coomassie stain to correct for differing protein amounts, following which the phosphorylation by WT KdpD was

set to 100%. Bars indicate the mean and error bars denote the standard deviation from biological triplicates (n = 3). Full phosphoprotein and Coomassie stains are shown in Supplementary Fig. 4. **b** Structural model of $KdpD_{NTD}$ in complex with the KdpB A domain by AlphaFold[34,35], with ATP modeled into the binding pocket of $KdpD_{NTD}$ by structural homology with the AAA+ ATPase Rix7 (PDB 6MAT[75]). **c** Kinase active site in the interaction model, with residues involved in ATP binding or phospho-transfer catalysis highlighted. **d** Suggested ASK-type mechanism for phosphorylation of $KdpB_{S162}$ by KdpD. Conservation of mutated residues and tests for autophosphorylation or ATP hydrolysis by $KdpD_{NTD}$ are shown in Supplementary Figs. 5, 6, respectively.

A/B-containing ATPases[36,37]. More precisely, the aspartate of the Walker B motif polarizes the gamma phosphate via coordination of a $Mg^{2+}$, while the glutamate polarizes a water molecule for hydrolysis; mutation of these residues in ATPases does not abolish ATP binding but prevents hydrolysis[37]. In the interaction model of $KdpD_{NTD}$ and the KdpB A domain, the Walker B motif ($KdpD_{D109/E110}$) is positioned ideally for a polarizing function in the serine kinase mechanism, with $KdpD_{E110}$ activating $KdpB_{S162}$ rather than a water. Indeed, mutating the residues of the Walker B motif individually ($KdpD_{D109A}$ and $KdpD_{E110A}$) abolished the phosphorylation of KdpB, indicating that both are essential for the phosphorylation mechanism. Interestingly, glutamate $KdpB_{E161}$ is positioned behind $KdpD_{E110}$ of the Walker B motif in the complex model, suggesting a supporting role in the polarization. Indeed, mutation $KdpB_{E161Q}$ resulted in a reduction in phosphorylation, but only by 64%. Based on its position behind $KdpD_{E110}$, it appears that $KdpB_{E161}$ potentiates the polarization of $KdpB_{S162}$ by the Walker B motif. These results suggest that $KdpB_{S162}$ directly attacks the γ-phosphate in a mechanism reminiscent of other ASKs, while the Walker B motif functions as in ATPases, with $KdpD_{E110}$ activating the serine for the nucleophilic attack, supported by $KdpB_{E161}$ (Fig. 3d).

To test the specificity of the ASK-type serine phosphorylation, $KdpB_{S162}$ was mutated to threonine and tyrosine, both of which resulted in no phosphorylation (Fig. 3 a). The identical observation for $KdpB_{S162Y}$ and $KdpB_{S162T}$ suggests that the effect is most likely due to active site geometry, in which only serine is suited for polarization by $KdpD_{E110}$ and $KdpB_{E161}$.

Further, we tested $KdpD_{NTD}$ for autophosphorylation and ATPase activity (Supplementary Fig. 6). In agreement with an ASK mechanism, in which neither of these occur, $KdpD_{NTD}$ is not phosphorylated under any conditions, even under non-steady-state conditions where phospho-transfer to KdpB is prevented by the mutation $KdpB_{S162A}$, and showed no ATPase activity, indicating that the Walker B motif does not exert a hydrolysis function as in Walker A/B-containing ATPases[36,37]. These results align with the mechanism deduced from the mutagenesis studies, supporting the characterization of the KdpD domain as an ASK.

## KdpD's ASK activity is conserved across species but distinctly regulated

After identifying the ASK activity of the KdpD domain in *E. coli* KdpD, we wondered how conserved this regulatory mechanism was across prokaryotic species. To investigate this, we analyzed the sequences of KdpD from 5495 species that feature an annotation for the Kdp pump in the UniProt database (Supplementary Data 1, 2). The dataset was initially based on that of a previous study[38], which was subsequently expanded to include all species from the LPSN database fitting the parameters[39]. KdpD sequences were classified according to their domain architecture, with particular emphasis on the presence of the KdpD and DHpCA domains, which confer the ASK and HK activities, respectively (Fig. 4a). Notably, this dataset is dependent on annotations and sequences in the Uniprot database. While faulty annotations were removed by sequence alignments, incomplete sequences cannot be excluded. However, the frequency of the five most common classes of KdpD in the dataset suggests that a sequencing artifact is, in these cases, highly unlikely. While almost 75% of species contain a 'full-length' KdpD version, featuring all the domains present in *E. coli* KdpD, a significant fraction deviates from this architecture, with the most significant versions shown in Fig. 4b and the distributions for individual phyla shown in Supplementary Fig. 7a, b. Some species feature other KdpD variants in addition to a 'full-length' version; in these cases, only the full-length variant was included in the analysis, as it fulfills the requirements for both HK and ASK activities. Interestingly, some species contain the ASK and HK activities on separate polypeptide chains, with the most common combination consisting of a KdpD/Usp domain ASK and a TM/GAF/DHpCA domain HK. Moreover, there are

multiple versions of KdpD in which the requirements for the HK are not fulfilled but the ASK domain is conserved. These can be separated into versions consisting of only the N-terminal KdpD and Usp domains, or versions in which the C-terminal DHpCA domains are truncated, removing the HK activity. In these species, it appears that the sole role of KdpD is the post-translational regulation of KdpFABC via the ASK-mediated phosphorylation. Indeed, KdpD versions lacking the domains for HK activity are predominantly found in species also missing the transcription factor KdpE; they make up almost 90% of KdpD versions found in these species. By contrast, species containing KdpE overwhelmingly feature KdpD versions or combinations that mediate a HK activity for transcriptional control in addition to the ASK activity (95%; Fig. 4c). Importantly, in almost all versions, the KdpD domain with the ASK function was preserved.

The shortened KdpD/Usp domain version poses an interesting conundrum, as all previously suggested $K^+$ sensing sites in *E. coli* KdpD (*Ec*KdpD), albeit only characterized for the HK, are located in the TM/GAF domains[7]. To investigate how post-translational regulation by KdpD is linked to $K^+$ levels in these species, we compared the stimulation of ASK activity between *E. coli* KdpD, the well-studied full-length chain, and a shorter KdpD/Usp domain version from *Deinococcus geothermalis* (*Dg*KdpD, UniProt accession Q1J2K3) (Fig. 5a). In vivo, the phosphorylation of $KdpB_{S162}$ in *E. coli* and the HK and phosphatase activities of KdpD are dependent on $K^+$ levels[12,14]. However, when purified in DDM, the HK activity is constantly low, while the phosphatase activity is high[10]. In agreement with this observation, KdpD solubilized in DDM here showed a constantly high serine kinase activity (Fig. 5b). Since a membrane environment was shown to restore the $K^+$ dependence of KdpD's HK activity[10], we solubilized KdpD directly from *E. coli* membranes in SMALPs to retain the $K^+$ stimulation of the serine kinase activity. Contrary to detergents like DDM, which displace lipids to form micelles tightly around the TM domain, SMALPs can form stable lipid discs containing the membrane protein. Indeed, KdpD's ASK activity in SMALPs was stimulated upon the addition of $K^+$, suggesting that the N-terminal domain also underlies the control of the $K^+$-sensing domains of KdpD (Fig. 5b). Conversely, *Dg*KdpD's ASK activity is not stimulated by $K^+$ (Fig. 5c). In *Staphylococcus aureus* KdpD, the Usp domain was reported to bind c-di-AMP, a second messenger, which in general has been implicated in a wide range of osmotic and $K^+$ homeostasis regulation mechanisms[40]. Binding of c-di-AMP led to the downregulation of *kdpFABC* expression[41,42]. Consequently, we suspected that c-di-AMP binding to the Usp domain could be the signal for ASK stimulation in KdpD versions lacking $K^+$ sensing domains. Indeed, phosphorylation of *E. coli* KdpB by *Dg*KdpD was stimulated in the presence of c-di-AMP, indicating that, here, the $K^+$-sensing role of KdpD is outsourced to the diadenylate cyclase that forms c-di-AMP at increased $K^+$ levels[43]. In agreement with this hypothesis, the majority of species featuring a shorter KdpD/Usp domain architecture also express a diadenylate cyclase (Fig. 4d). At the same time, the fact that *Dg*KdpD can phosphorylate *E. coli* KdpB shows that the regulatory phosphorylation is conserved, independent of the input domains.

## Discussion

The regulation of KdpFABC by phosphorylation of $KdpB_{S162}$ is an essential process in facilitating microbial adaptability to changing $K^+$ conditions. Here, we show that the rapid, irreversible inhibition of KdpFABC at high $K^+$ is widely conserved and mediated directly by a Walker A/B motif in the N-terminal KdpD domain of KdpD in an atypical serine kinase mechanism, making KdpD a tandem serine histidine kinase. Moreover, we identified a shorter form of KdpD without HK domains in some species that only fulfills the role of post-translational control of KdpFABC, regulated by the second messenger c-di-AMP.

Physiologically, the use of KdpD to also inhibit KdpFABC post-translationally in *E. coli* is remarkably smart since, this way, two outcomes, HK and ASK activities, are controlled by the same input: $K^+$

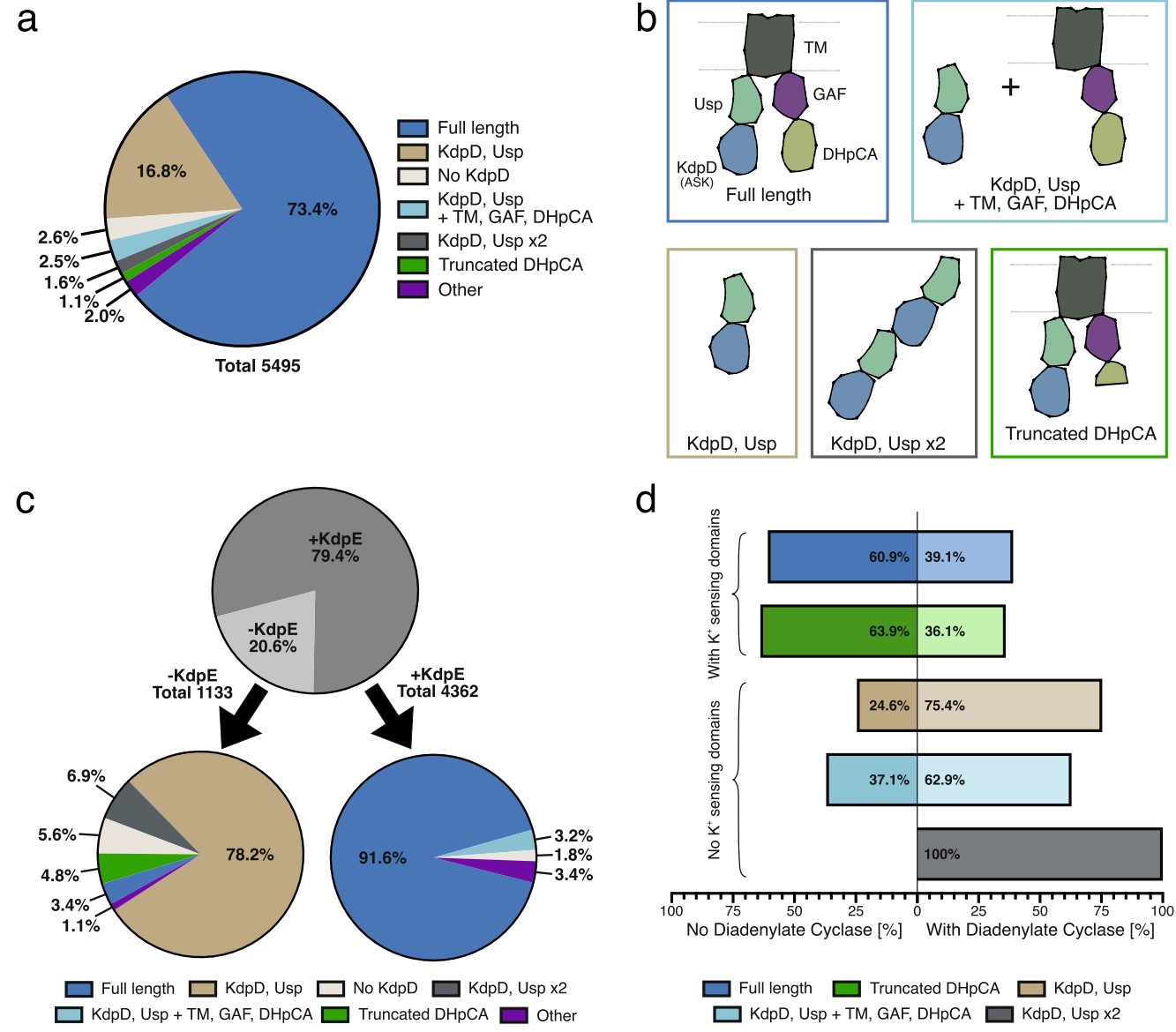

**Fig. 4 | Domain variability of KdpD from different species.** The domain organization of KdpD sequences from 5495 species featuring annotated Kdp(F)ABC in the UniProt database was analysed (see Supplementary Fig. 7). **a** Distribution of KdpD versions in the analyzed set of sequences. **b** Schematic domain organization of the five most common versions of KdpD in the analyzed set of sequences. The ASK activity requires the N-terminal KdpD domain, HK activity is mediated by the C-terminal DHpCA domain. **c** Distribution of KdpD versions in the analyzed set of sequences in species with or without the response regulator KdpE. Over 95% of KdpD in species without KdpE lack the domains for an HK activity, while close to 95% of KdpD in species with KdpE also contain an HK domain. **d** Correlation of KdpD versions with or without the K⁺-sensing TM and GAF domains to the presence of a diadenylate cyclase.

sensing by the tandem serine histidine kinase. Thus, KdpD acts as a single, centralized regulator for both KdpFABC activity and gene expression to ensure its required strict control[6,12]: At low K⁺ conditions, KdpD's HK activity is stimulated to phosphorylate $KdpE_{D52}$ via a phospho-histidine intermediate in the DHp domain, causing KdpE to dimerize and activate *kdpFABC* transcription[5]. The activation of the histidine kinase is further supported by the binding of UspC to the Usp domain[44] (Fig. 6a). Conversely, at moderate to high K⁺ concentrations, K⁺ binds to KdpD's sensing domains in the TM (extracellular) or GAF domains (intracellular), switching the DHp domain from its histidine kinase to its phosphatase activity and mediating the dephosphorylation of KdpE[6–9] (Fig. 6b). Simultaneously, at these conditions, the N-terminal KdpD domain binds ATP[32], and, as we now know, phosphorylates $KdpB_{S162}$ as an ASK to inhibit the pump on a post-translational level to prevent excessive K⁺ uptake. Specificity for the reaction is probably provided by $KdpB_{E161}$, which is required for the

polarization of $KdpB_{S162}$. This could actually explain how the ASK prevents unspecific hydrolysis of ATP by the Walker A/B motif that, in other ATPases, is able to independently hydrolyze ATP: only in the complex with the substrate is the polarization by the Walker B motif of KdpD strong enough to facilitate ATP cleavage.

The direct phospho-transfer from KdpD to $KdpB_{S162}$ presented here coincides with multiple previously reported observations: KdpD was shown to interact with KdpB, but for an unknown reason[45,46]. Both KdpFABC and KdpD are also regulated by cardiolipin, suggesting a possible co-localization in the membrane[15,47,48]. Moreover, the KdpD domain containing the Walker A/B motif is highly conserved among KdpDs from different species, but not found in other sensor kinases[31]. Thus, it provides an ideal platform for a specific interaction with KdpB, which is a prerequisite for a precise and selective regulatory function.

Species featuring a short KdpD with only the N-terminal KdpD and Usp domains, like *Dg*KdpD, appear to employ a more centralized

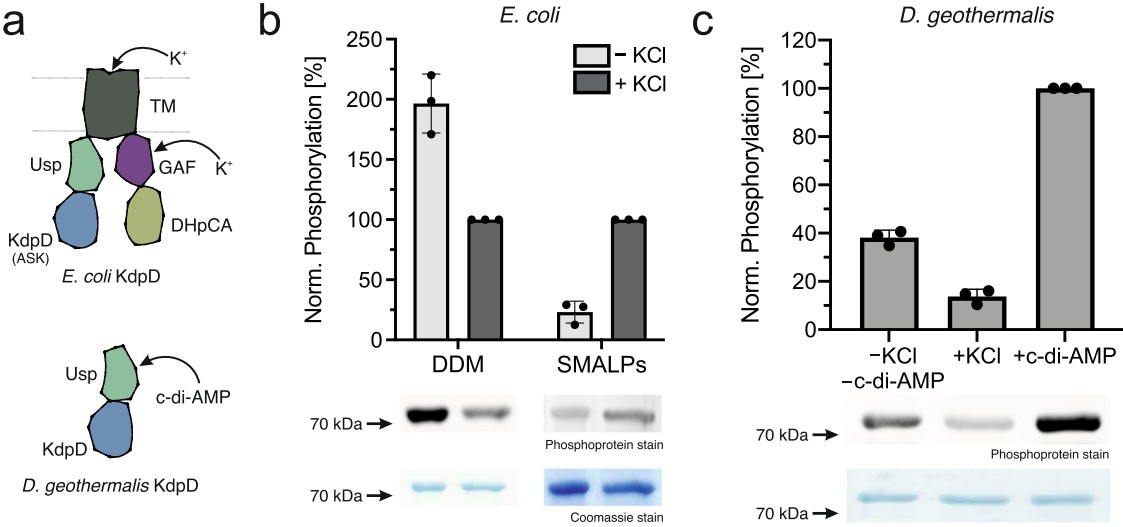

**Fig. 5 | Control of E. coli and D. geothermalis KdpD ASK activity by K⁺ and c-di-AMP. a** Schematic domain structure of *Ec*KdpD and *Dg*KdpD. Stimulation of ASK activity of *Ec*KdpD solubilized in DDM or SMALPs, which mimic the native membrane environment, for *Ec*KdpFABC by 400 mM KCl (**b**), and *Dg*KdpD ASK activity for *Ec*KdpFABC by 400 mM KCl or 1 mM c-di-AMP (**c**). Phosphoprotein gel stain of purified KdpFABC separated by SDS-PAGE, indicating the phosphorylation state of KdpB. The corresponding Coomassie stain shows the total protein amount loaded per lane. Phosphoprotein stains were quantified with ImageJ and normalized to the quantification of the Coomassie stain to correct for differing protein amounts, following which the phosphorylation in the presence of KCl in (**b**) or the presence of c-di-AMP in (**c**) was set to 100%. Bars indicate the mean and error bars denote the standard deviation from biological triplicates ($n = 3$). The full Phosphoprotein- and Coomassie-stained gels and the purifications of *Ec*KdpD in SMALPs and *Dg*KdpD are shown in Supplementary Fig. 8.

regulation: post-translational control of KdpFABC by c-di-AMP binding to KdpD (Fig. 6c, d). At low K⁺ levels, the concentration of c-di-AMP is low[43]; consequently, KdpD is inactive. At high K⁺ conditions, a diadenylate cyclase forms c-di-AMP, which binds to the Usp domain of KdpD and activates the ASK activity of the NTD to inhibit KdpFABC. In addition to the post-translational regulation, some species of the *Bacillota* phylum also use c-di-AMP for the transcriptional control of *kdpFABC* via a c-di-AMP-responsive *ydaO* riboswitch, putting the entire Kdp system under the control of the second messenger[38]. The full control of some Kdp systems by c-di-AMP is reminiscent of the regulation of K⁺ transport systems KtrAB and KimA in *Bacillus subtilis*, whose expression is controlled by c-di-AMP binding to the *ydaO* riboswitch and which are additionally allosterically regulated by the second messenger[43,49–53]. This works highlights another role for c-di-AMP in K⁺ homeostasis, emphasizing the importance of the second messenger in bacterial adaptability. But what about organisms featuring neither 'full-length' KdpD nor a riboswitch? In some species such as *Deinococcus radiodurans, Anabaena sp.,* and *Alicyclobacillus acidocaldarius, kdp* expression is regulated by a separate TCS[54–56]. Interestingly, in these species, the ASK-containing chain is retained downstream of the pump in the *kdp* operon, while the HK is separate. This indicates that the read-through expression of the ASK is important to ensure sufficient protein levels to efficiently inhibit the pump when required. In other organisms, no separate HK was identified, and the expression of *kdpFABC* could be constitutive or controlled by another unknown mechanism.

What remains to be addressed is why KdpD is controlled by different inputs and whether one is superior over the other. The regulation by K⁺ levels could allow for a more independent control of KdpFABC from other K⁺ transport systems. However, c-di-AMP binding sites from different proteins show large structural differences, so a graded control of K⁺ transport systems could also be facilitated by different affinities for c-di-AMP. A particularly intriguing question is how KdpDs that harbor both the K⁺ sensing domains and the c-di-AMP binding site, like for example *S. aureus* KdpD[41], are regulated. Are the different stimuli additive or is one dominant over the other? Or are the different functions of KdpD controlled separately, i.e. is the ASK

activity controlled by c-di-AMP and the HK activity by K⁺ sensing? In *Ec*KdpD, multiple signals regulate the HK activity, including the two K⁺ sensing domains and UspC binding in the Usp domain. The integration of multiple signals bears the advantage of an even more differentiated regulation of the sensor kinase. However, control of the ASK activity remains to be further characterized

What generally surprised us after sequence analysis of KdpDs from different species was the high variability in the types of KdpD found, even if the dataset is limited by the number of sequenced genomes and the annotation of sequences in the UniProt database. The distribution of KdpD versions in different phyla makes speculation about the evolution of the KdpD(E) regulatory system enticing (Supplementary Fig. 7). In some species, most commonly in *Acidobacteriota* and *Chloroflexota*, ASK and HK activities are separated on two polypeptide chains, suggesting that these functions evolved separately with distinct regulatory mechanisms. The HK chains contain the domains known to be K⁺ sensors in *E. coli* KdpD, suggesting that the HK is always controlled by K⁺. In these species, the shorter ASK KdpD is controlled by the Usp domain. The phyla featuring mainly this short KdpD version (*Bacteroidota, Cyanobacteriota, Deinococcota, Myxococcota*) mostly contain diadenylate cyclases, suggesting that control by c-di-AMP, as shown here for *D. geothermalis* KdpD, could be a common theme. The ASK and HK chains must have at some point been combined into the *E. coli*-like 'full-length' chain by gene fusion, after which ASK activity came under the control of the K⁺-sensing domains. This is supported by the fact that some 'full-length' KdpDs still bind c-di-AMP in their Usp domain[41,57]. Fascinatingly, some phyla (*Campylobacterota, Archaea*) appear to have evolved back to lose the HK activity, featuring a truncated DHpCA domain. These phyla are often found in extreme conditions, where *kdpFABC* expression may be constantly needed, making transcriptional control superfluous. Accordingly, a significant fraction of *Archaea* completely lacks any form of KdpD. The phylum with the highest KdpD variance is *Bacillota*, probably due to their natural competence, which allowed them to scavenge *kdpD* genes from different sources in their surroundings[58].

Finally, we would like to place the tandem serine histidine kinase activity of KdpD into the context of non-canonical functions of other

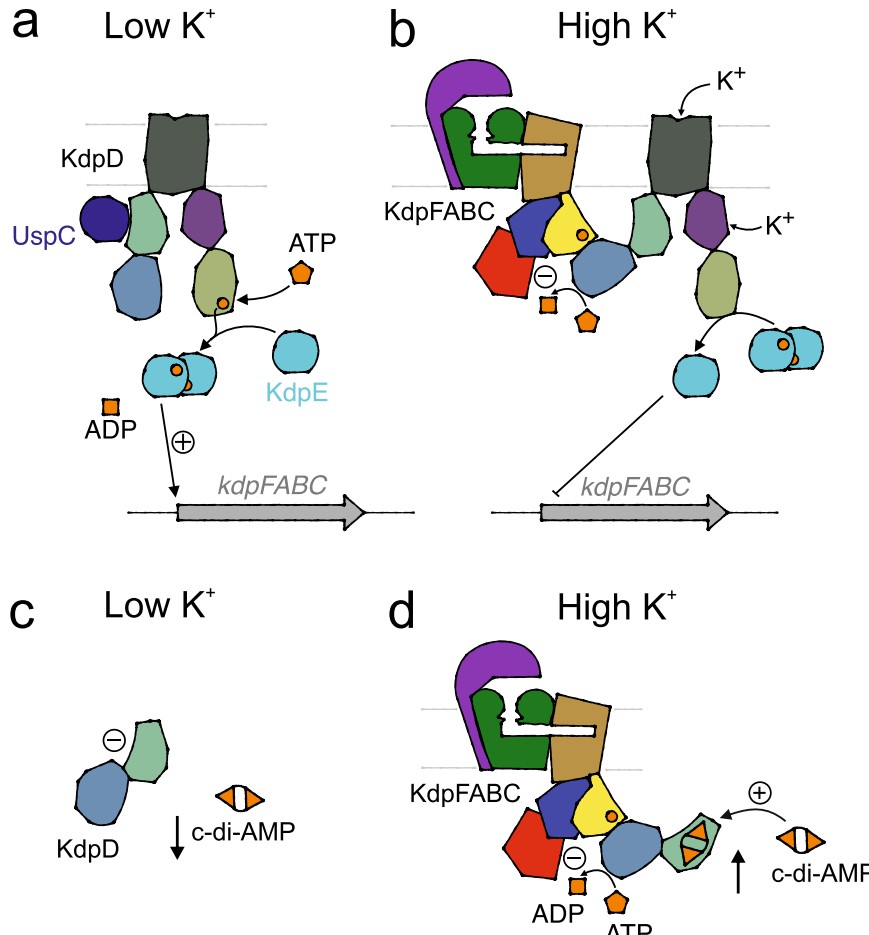

**Fig. 6 | Regulation strategies of the Kdp system. a, b** Transcriptional and post-translational regulation of KdpFABC in *E. coli* by KdpD, which senses K⁺. **a** Activation of KdpD's histidine kinase at low K⁺ conditions. At osmotic stress, UspC binds to the Usp domain, aiding activation. **b** Activation of KdpD's cognate phosphatase and atypical serine kinase at high K⁺ conditions. **c, d** Regulation of KdpFABC by c-di-AMP-sensitive short KdpD. **c** At low K⁺ conditions, cytosolic c-di-AMP concentrations are low, rendering KdpD inactive and allowing K⁺ uptake by KdpFABC. **d** At high K⁺ levels, c-di-AMP accumulates in the cytosol. The second messenger binds to the Usp domain of KdpD, activating the serine kinase activity for KdpB to inhibit further K⁺ uptake.

sensor kinase systems. Canonically, two-component systems transduce environmental signals downstream to induce a physiological response on the transcriptional level. However, recently, this view of sensor kinases as a linear signaling pathway has shifted, as multiple cases of transporter-kinase interactions have been characterized[59–61]. In these systems, the cognate transporter interacts with the sensor kinase to convey additional stimuli. The additional regulation can range from feedback control on transcription levels[60] to detection of transport activity[62–64] or even sensing of stimuli by the transporter[65]. However, in all of the described systems, the interactions serve to provide additional stimuli to the kinase in a form of extended feedback regulation. The regulation of KdpFABC by KdpD additionally shows post-translational control of a transporter by its sensor kinase. However, the post-translational regulatory role could be more widespread, as the presence of Walker A/B motifs in a cytosolic domain has been described for other histidine kinases[66,67].

The data presented here shows that the inhibition of KdpFABC is mediated by an atypical serine kinase domain in the N terminus of KdpD, completing the puzzle of how cells tightly regulate K⁺ uptake by KdpFABC to ensure robust K⁺ homeostasis in rapidly changing conditions. Further work will be required to elucidate the exact sensing and intramolecular signal transduction mechanisms that allow KdpD to dynamically regulate its serine kinase activity. The presence of an atypical serine kinase domain on a histidine kinase opens up new possibilities in the search for bacterial kinases: domains previously

thought to be ATPases or regulatory nucleotide binding sites may have unrecognized kinase functions, helping to explain how the diverse phosphorylation patterns in bacteria can be facilitated.

## Methods

### Cloning and protein production

*Escherichia coli kdpFABC, kdpD*, and *kdpD_NTD* and *Deinococcus geothermalis kdpD* were cloned into the expression vector pBXC3H by FX cloning[68]. Point mutations based on these constructs were generated by site-directed mutagenesis[69]. To clone *E. coli kdpD* or *kdpD_ΔDHpCA* into pBAD33, the gene or gene fragment was amplified from genomic DNA with 5' and 3' restriction sites for XbaI and PstI, respectively. The purified PCR product and pBAD33 were digested with XbaI and PstI according to the manufacturer's instructions (Thermo Fisher Scientific; Waltham, MA), and the resulting fragments mixed in a molar ratio of 5:1 for ligation with T4 DNA ligase according to the manufacturer's instructions (New England Biolabs; Ipswich, MA). Point mutations based on these constructs were generated by site-directed mutagenesis. For cloning of GFP-tagged *kdpD* constructs in pBAD33, pBAD33 with the untagged *kdpD* gene was amplified with primers binding at the 3' end of the ORF, omitting the stop codon and containing a PstI or HindIII (reverse/forward primer, respectively) restriction site. *sfGFP* was amplified with 5' PstI and 3' HindIII restriction sites. PCR products were digested with PstI and HindIII (Thermo Fisher Scientific; Waltham, MA) before ligation with T4 DNA ligase. All *kdpFABC* and *kdpD*

constructs used are listed in Supplementary Table 1 and primers used are listed in Supplementary Table 2.

KdpFABC variants were produced in *E. coli* LB2003 (F⁻ *thi metE rpsL gal rha kup1 ΔkdpFABC5 ΔtrkA*)[70] or TK2281 (*ΔkdpFABCDE trkA405 trkD1 nagA thi rha lacZ*)[71] cells transformed with the respective pBXC3H derivatives. For co-expression with *E. coli kdpD* constructs, cells were co-transformed with an orthogonal pBAD33 derivative encoding the KdpD variant. Cells were grown in 4 l KML with 100 µg/ml ampicillin (and 30 µg/ml chloramphenicol when co-transformed with pBAD33 derivatives). Cultures were inoculated to an $OD_{600}$ of 0.1 and induced with 0.002% L-arabinose at $OD_{600}$ 1.3 for one hour at 37 °C.

*E. coli* KdpD variants were produced in *E. coli* TK2281 cells. Cells transformed with the respective pBXC3H derivatives encoding the KdpD variants were grown in 6 l KML with 100 µg/ml ampicillin. Cultures were inoculated to an $OD_{600}$ of 0.1 and induced with 0.004% L-arabinose at $OD_{600}$ 0.9 for one hour at 37 °C.

*D. geothermalis* KdpD variants were produced in *E. coli* TK2281 cells. Cells transformed with the respective pBXC3H derivatives encoding the KdpD variants were grown in 12 l KML with 100 µg/ml ampicillin. Cultures were inoculated to an $OD_{600}$ of 0.1, grown to an $OD_{600}$ of 0.7 at 37 °C, and induced with 0.1% L-arabinose at 30 °C for two hours.

Bacterial strains and plasmids are available from the Hänelt group upon request.

## KdpFABC purification

KdpFABC variants were purified as previously described, omitting size exclusion chromatography after anion exchange chromatography[27,28]. In short, after cell disruption using a Stansted cell disruptor (Homogenising Systems Ltd.; Essex, U.K.), membranes were harvested and incubated with 1% (w/v) DDM overnight. Unsolubilized proteins were removed by centrifugation at $150,000 \times g$ for 30 min, and the supernatant incubated with 2 ml Ni²⁺-NTA sepharose pre-equilibrated with KdpFABC Ni-NTA buffer (50 mM Tris-HCl pH 7.5, 20 mM MgCl₂, 150 mM NaCl, 10% glycerol, 0.025% w/v DDM) and 10 mM imidazole for 1 h. After washing with 25 column volumes of KdpFABC Ni-NTA buffer at an increased imidazole concentration of 50 mM, the protein-bound Ni²⁺-NTA was subjected to an additional low-salt washing step of 25 column volumes (10 mM Tris-HCl pH 7.5, 10 mM MgCl₂, 10 mM NaCl, 50 mM imidazole, 0.025% w/v DDM), before elution with a low-salt imidazole elution buffer (10 mM Tris-HCl pH 7.5, 10 mM MgCl₂, 10 mM NaCl, 250 mM imidazole, 0.025% w/v DDM). Elution fractions were subjected to AIEX using AIEX buffer (10 mM Tris-HCl pH 8.0, 10 mM MgCl₂, 10 mM NaCl) and a HiTrap Q HP column (Cytiva; Marlborough, MA, USA) and eluted with a NaCl gradient from 10-500 mM.

## *E. coli* KdpD purification

KdpD variants were purified largely as previously described[10]. In short, after cell disruption using a Stansted cell disruptor (Homogenising Systems Ltd.; Essex, U.K.), membranes were harvested and washed twice in a low-salt EDTA buffer (1 mM Tris-HCl pH 7.5, 3 mM EDTA). 10 ml aliquots of membranes set to a total protein concentration of 10 mg/ml were supplemented with 600 mM NaCl and 2% (w/v) LDAO and incubated for 30 min at 4 °C. Unsolubilized proteins were removed by centrifugation at $150,000 \times g$ for 45 min, and the supernatant incubated with 2 ml Ni²⁺-NTA sepharose pre-equilibrated with KdpD Ni-NTA buffer (50 mM Tris-HCl pH 7.5, 500 mM NaCl, 10% glycerol, 10 mM β-mercaptoethanol, 0.04% w/v DDM) and 10 mM imidazole for 30 min. After washing with 25 column volumes KdpD Ni-NTA buffer at an increased imidazole concentration of 50 mM, the protein-bound Ni²⁺-NTA was subjected to an additional low-salt washing step of 25 column volumes (50 mM Tris-HCl pH 7.5, 10% v/v glycerol, 10 mM β-mercaptoethanol, 50 mM imidazole, 0.04% w/v DDM), before elution with a low-salt imidazole elution buffer (50 mM Tris-HCl pH 7.5, 10% v/v glycerol, 10 mM β-mercaptoethanol, 400 mM imidazole, 0.04% w/v DDM). Elution fractions were subjected to AIEX using using AIEX buffer (50 mM Tris-HCl pH 7.5, 10 mM β-mercaptoethanol, 0.04% w/v DDM) and a HiTrap Q HP column (Cytiva; Marlborough, MA, USA) and eluted with a NaCl gradient from 0 to 800 mM.

KdpD$_{NTD}$ was purified in the same manner, incubating the cytosol with Ni²⁺-NTA and omitting detergent from all buffers.

For purification of KdpD in SMALPs, KdpD-containing membranes were prepared as described above. 10 ml aliquots of membranes set to a total protein concentration of 10 mg/ml were supplemented with 4% (w/v) SMALP 140 (Orbiscope; Geleen, the Netherlands), set to a total volume of 20 ml, and incubated for 18 h at 4 °C. Unsolubilized proteins were removed by centrifugation at $150,000 \times g$ for 45 min, and the supernatant diluted tenfold with SMALP buffer (50 mM Tris-HCl pH 7.5, 10 mM β-mercaptoethanol) supplemented with 500 mM NaCl before being incubated with 2 ml Ni²⁺-NTA sepharose pre-equilibrated with KdpD Ni-NTA buffer, omitting DDM, and 10 mM imidazole for 18 h. After washing with 25 column volumes KdpD Ni-NTA buffer, omitting DDM, at an increased imidazole concentration of 50 mM, the protein-bound Ni²⁺-NTA was subjected to an additional low-salt washing step of 25 column volumes (50 mM Tris-HCl pH 7.5, 10 mM β-mercaptoethanol, 50 mM imidazole), before elution with a low-salt imidazole elution buffer (50 mM Tris-HCl pH 7.5, 10 mM β-mercaptoethanol, 400 mM imidazole). Elution fractions were subjected to SDS-PAGE to determine fractions containing KdpD, which were subsequently pooled and buffer exchanged to SMALP buffer with a ZebaSpin™ 7 K MWCO desalting column (ThermoFisher Scientific, Waltham, MA).

## *D. geothermalis* KdpD purification

*D. geothermalis* KdpD was purified largely like *E. coli* KdpD$_{NTD}$, omitting the final AIEX chromatography step. In short, after cell disruption using a Stansted cell disruptor (Homogenising Systems Ltd.; Essex, U.K.), the cytosol was supplemented with 10 mM imidazole and 500 mM NaCl and incubated with 6 ml pre-equilibrated Ni²⁺-NTA sepharose for 1 h. After washing with 50 column volumes of wash buffer (50 mM Tris-HCl pH 7.5, 500 mM NaCl, 5% w/v glycerol, 10 mM β-mercaptoethanol, 50 mM imidazole), the column was incubated for 1 h with 1 mg/ml 3 C protease in elution buffer (50 mM Tris-HCl pH 7.5, 300 mM NaCl, 10% w/v glycerol, 10 mM β-mercaptoethanol) under light stirring before elution with elution buffer. Imidazole was removed from elution fractions by buffer exchange using a ZebaSpin™ 7 K MWCO desalting column (ThermoFisher Scientific, Waltham, MA).

## ATPase assay

ATP hydrolysis by purified KdpFABC variants was tested by malachite green ATPase assay[72]. In brief, each reaction contained 2 mM ATP and 1 mM KCl. The reaction was started by adding 0.5–1.0 µg protein, and carried out for 5 min at 37 °C.

## In vitro phosphorylation

To show phospho-transfer between KdpD and KdpFABC in vitro, a three-fold molar excess of purified KdpD was mixed with KdpFABC variants purified from *E. coli* TK2281 cells (i.e., without prior KdpB$_{S162}$ phosphorylation) and 400 mM KCl. All KdpFABC variants used featured the mutation KdpB$_{D307N}$ to preclude the catalytic phosphorylation of the P domain. The reaction was started by the addition of 5 mM ATP, and incubated for 30 min at room temperature, after which the reaction was stopped by mixing with SDS sample buffer for subsequent SDS-PAGE and phosphoprotein gel stain.

To test the phosphorylation state of KdpD$_{NTD}$, the same protocol was performed in the absence and presence of 5 mM ATP, and in the presence of KdpFAB$_{D307N}$C or KdpFAB$_{S162AC/D307N}$ in a 3:1 molar ratio KdpD$_{NTD}$:KdpFABC.

To test the stimulation of *E. coli* or *D. geothermalis* KdpD serine kinase activity, the same protocol was performed in the absence and presence of 400 mM KCl or 1 mM c-di-AMP.

## Phosphoprotein gel stain

For phosphoprotein gel staining, purified protein samples were separated by 12.5% SDS-PAGE[73]. Phosphorylated proteins were stained by Pro-Q™ Diamond Phosphoprotein Gel Stain (ThermoFisher Scientific, Waltham, MA) according to the manufacturer's instructions. Gels were subsequently stained with Coomassie brilliant blue R-250 to visualize the entirety of the loaded protein.

## Sequence analysis

KdpD sequences and UniProt accession numbers from species analyzed in a previous study[38] or included in the LPSN database[39] were extracted and analyzed for homology to *E. coli* KdpD, the presence of the KdpD domain Walker A and B motifs, and the presence of the HK catalytic histidine by pairwise alignment using Clustal Omega[74], following which they were manually classified based on domain structure and potential kinase functionality. For each species, the UniProt accession number for KdpE and a diadenylate cyclase was likewise extracted.

## Expression test

To control expression of *kdpD* constructs from pBAD33 derivatives in co-expression experiments, GFP-tagged variants of KdpD were produced in *E. coli* TK2281 cells co-transformed with pBXC3H-*kdpFABC*. Samples from each culture were set to an $OD_{600}$ of 30 in SDS sample buffer, and subjected to 12.5% SDS-PAGE with subsequent analysis of in-gel GFP fluorescence (Ex. 485 nm/Em. 535 nm).

## Statistics and reproducibility

All biochemical experiments were performed in biological triplicate, except for the ATPase assay presented in Supplementary Fig. 6b, which was performed in technical triplicate. All replicates were successful, and no data were excluded from the analyses. In Figures, bars represent the mean and error bars the standard deviation from triplicate measurements. No statistical method was used to predetermine sample size. The experiments were not randomized, and the investigators were not blinded to allocation during experiments and outcome assessment.

## Reporting summary

Further information on research design is available in the Nature Portfolio Reporting Summary linked to this article.

# Data availability

All information required to reanalyze the data reported in this paper has been deposited in a Source Data File and in Supplementary Data 1 and 2. The List of Prokaryotic names with Standing in Nomenclature can be found under https://www.bacterio.net/. UniProt Accession numbers used in the analysis of this work are listed in Supplementary Data 1. PDB 6MAT was used for structural analysis. Source data are provided with this article. Source data are provided with this paper.

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

## Acknowledgements

This work was funded by the State of Hesse in the LOEWE Schwerpunkt TRABITA, a DFG Heisenberg grant (HA6322/5-1), and funding from the Aventis Foundation and the Uniscientia Stiftung.

## Author contributions

J.M.S.: Conceptualization, Validation, Formal analysis, Investigation, Writing—original draft, Writing—review & editing, Visualization. S.K.: Investigation, Formal analysis, Writing—review & editing. P.J.N.B.: Investigation, Formal analysis, Writing—review & editing. K.J.: Investigation, Formal analysis. M.W.: Investigation. J.G.: Investigation. I.H.: Conceptualization, Writing—review & editing, Supervision, Project administration, Funding acquisition.

## Funding

## Competing interests

The authors declare no competing interest.
