## [Peer Review File · Nature Communications]

KdpD is a tandem serine histidine kinase that controls K⁺ pump KdpFABC transcriptionally and post-translationallyReviewer #1 (Remarks to the Author):

With great interest I was reading the manuscript by Silberberg and coworkers about the control of potassium uptake in the Gram-negative model bacterium *Escherichia coli*. *E. coli* possesses several uptake systems potassium, which is the most abundant cellular cation in many organisms. In *E. coli* as well as in other organisms, the rapid uptake of potassium ions is needed for the survival of the cell when it encounters hyperosmotic growth conditions. The *E. coli* KdpFABC potassium uptake system has a high affinity for the osmolyte. Therefore, the activity of the system needs to be tightly regulated to prevent the uptake of potassium ions to toxic levels. Since a while it is known that the KdpB subunit of the KdpFABC system can be phosphorylated. The phosphorylation of KdpB is required to shut down potassium uptake. However, the kinase responsible for the inactivation of the KdpFABC system was so far unknown.

By employing a series of elegant genetic and biochemical approaches Silberberg and coworkers demonstrated that the sensor kinase KdpD of the KdpDE TCS (that also controls the expression of the *kdp* genes) can phosphorylate KdpB, thereby inactivating the KdpFABC system. Since a long-standing open question has been solved the work is a very important contribution to the field of osmoregulation.

I have only a few suggestions that should be addressed by the authors to improve the manuscript.

Major points

1. In my point of, the title of the manuscript is misleading. I would suggest to change the title as follows: "Sensory kinase KdpD is a tandem serine histidine kinase controlling K⁺ pump KdpFABC on the transcriptional and post-translational level". KdpD is required to activate the transcription of the *kdp* genes (transcriptional level regulation) and KdpD phosphorylates and inactivates the KdpFABC system (post-translational level regulation).

2. Phylogenetic analysis of the Kdp system. Comparative genomics can often lead to wrong conclusions due to sequencing errors and wrong annotations. For instance, it has been wrongly described that in some organisms the *cdaAR* genes are fused to one gene encoding a longer protein that harbors a diadenylate cyclase (DAC) domain and the regulator of the DAC. Due to sequencing errors this assumption is wrong. Did the authors consider this problem when analysing the sequences of KdpD orthologs?

3. Discussion section: The tight regulation of the KdpFABC system at the transcriptional and post-translational level strongly resembles the situation in *Bacillus subtilis* and other related species. In *B. subtilis*, the KtrAB system is regulated at the transcriptional and post-translational level by the second messenger c-di-AMP. The authors might consider mentioning this in the discussion.

Minor points

Page 1, line 29: The Usp domain is not explained in the abstract. Please either explain or delete.

Page 2, lines 47 and 50: better "kdpFABC".

Page 3, line 84: better "that controls the transcription of the *kdpFABC* genes and directly phosphorylates..."

Page 6, figure 2: The gel figure seems to be cut at the left site. Please explain!

Page 11, line 251: better "indicating that the Walker B motif..."

Page 13, line 308: "a second messenger, which..."

Page 13, line 314: Please provide a reference here.

Page 15, line 351: better "ASK"

Page 28, line 668: "B. Macek"

Reviewer #2 (Remarks to the Author):

K⁺ homeostasis plays a major role in the maintenance of bacterial osmolality. As a high-affinity K⁺ transport system, the expression of KdpFABC is tightly regulated by a two-component system, KdpDE. In this manuscript, the authors found that KdpD, a previously described HK of the TCS KdpDE, additionally mediates an inhibitory serine phosphorylation of KdpFABC at high K⁺ levels, using not its C-terminal HK domain but an atypical N-terminal serine kinase (ASK) domain. This finding reveals a new regulatory mechanism in the K⁺ transport system: at high K⁺ concentrations, KdpD not only repressed kdpFABC expression at the transcriptional level, but also irreversibly inhibited KdpFABC transport at the post-translational level by phosphorylating KdpB. They also analyzed and compared the structure of KdpD in bacteria and revealed different regulation of KdpD, e.g., in *E. coli*, the full-length KdpD is controlled by the concentration of K⁺, whereas in *Deinococcus geotherm*, the kdpD, which contains only USP and ASK domain, can be regulated by c-di-AMP, which affects the phosphorylation of KdpB. The finding adds to the diversity of sensor kinase functions and expand the framework of bacterial K⁺ regulatory mechanisms.

I have some comments listed below

1. The rapid, irreversible inhibition of KdpFABC at high K⁺ is widely conserved and mediated directly by the KdpD's ASK domain. Since there is no covalent bond formation between KdpFABC and KdpD ASK domain, why the inhibition by ASK is irreversible?
2. In Fig. 5, DDM or SMALPs were used to mimic the native membrane environment and for EckdpD or DgKdpD activity assay. Yet, in Fig. 5b, when DDM was used to mimic membrane environment, the phosphorylation activity of EckdpFABC seems to be attenuated in the presence of 400 mM K⁺, not simulated as described.
3. Also, in Fig. 5, DDM was used to mimic the native membrane environment. Does DDM forms a micelle in this case? When EckdpD is reconstituted into this micelle, how can one control its orientation? Is the KdpD/Usd domains exposed to water or buried inside the micelle? Can K⁺ penetrate the micelle to control the KdpD ASK activity if they are buried inside the micelle?
4. Also, in Fig. 5, the DgKdpD system is water soluble, thus no DDM or SMALPs is required to do the assay. But again, the K⁺ seem to substantially attenuate the phosphorylation activity of DgKdpD. Where does K⁺ bind to affect this activity?
5. In Fig. 6c, why does the addition of K⁺ make KdpB phosphorylation decrease? Are there other pathways by which K⁺ affect KdpB phosphorylation?
6. The authors found that the N-terminal KdpD domain of KdpD is an atypical serine kinase (ASK) domain, but it is not labeled in any figure?
7. Please describe in detail how DgKdpD is regulated by c-di-AMP?
8. An evolutionary tree should be drawn to elucidate the functional differentiation of KdpD protein.
9. Typo: line 92: controlled either by directly by the K⁺ concentration or by
10. Supplementary Fig. 6c (line 316) is not visible.
11. Please briefly explain DDM and SMALP in the text.

Reviewer #3 (Remarks to the Author):

This manuscript by Silberberg et al. describes a very interesting dual activity borne by a His-Kinase, KdpD, previously known to be part of a two-component systems which controls the expression of a K⁺ pump, KdpFABC. Here, it is shown that in addition to the classical His-kinase function, KdpD is also capable to phosphorylate the Ser162 residue of the KdpB subunit of the K⁺ pump by an atypical serine kinase (ASK) to regulate post-transcriptionally the function of the pump. This ASK activity is borne on a N-terminal domain of the KdpD and in some bacterial species, this domain is only fused to a Usp domain, no HK domain is therefore present. In such a case, instead of being controlled by the K⁺ level, as in full-length KdpD, the ASK can be controlled by a second messenger such as cyclic di-AMP, as shown here from the *Deinococcus geothermalis* enzyme. This result clearly expands the knowledge on how the function of a K⁺ pump is optimally controlled both at the translational and post-transcriptional level by a dual kinase, sometimes borne on different polypeptide, and that this regulation is finely tuned by the use of different intra- or extra-cellular signals.

Overall, the experiments showing the new ASK activity are well-performed and quite convincing and the results reported clearly warrant to be published.

However, I have a major concern regarding the lack of ATPase activity reported for the ASK domain. The amount of protein used (0.5-1 microgr.) is really too low if one wants to detect a low level of ATPase activity. It has been reported that some protein kinases do exhibit a very low level of ATPase activity when no protein substrate is provided in vitro (e.g. 10h-1, please see Nguyen et al. *J Mol Biol.* 2017 Oct 13;429(20):3056-3074. doi: 10.1016/j.jmb.2017.08.016.). This does make sense as some water molecules near the catalytic site can promote this low level of ATPase activity in the absence of a protein substrate. So before ruling out any ATPase activity of the ASK domain, the authors need to perform the ATPase assay using much more enzyme (e.g. 20 ug) and with a longer incubation time (e.g. 30 minutes at 37°C).

Also, many Ser/Thr/Tyr kinases, even aspecific ones, are capable to use classical 'in vitro' kinase substrates such as the myelin basic protein. So it would be nice that the authors try to phosphorylate such a kinase substrate in vitro using full-length KdpD and the catalytic ASK domain. This will nicely corroborate the Ser kinase activity of KdpD.

Minor remarks:

-Line 92 : "by directly by the": remove the first 'by'.

-Fig. 2. Please show the membrane limit on the scheme, color code the names of the subunit according to the figure, and use different symbol for the D307 and S162. The activity is normalized to 100% but indicate the specific activity in the legend

-Why do you need to use a triple mutant KdpDG36A/K37A/T38C in the Walker A motif, referred to as KdpDΔWalkerA? Usually, a point mutation of the Lysine of the WA is quite enough to kill the activity ?

-Please, show as Supplemental data an alignment of KdpD highlighting the Walker A and B motifs and all the other residues mutated afterwards.

-Line 158 : Ref 32 is written in parenthesis.

- Line 268: the ASK and HK activities can be borne on separate polypeptide chains: are they part of the same operon in this case?

- Fig 4. What the group 'others' stands for ? Are the sequences truly related to KdpD?

- Line 316 : There is no Supplementary Fig. 6c provided with the MS.

-Fig. 5: is it 5 mM c-di-AMP or 1 mM as mentioned in Supplemental data and in the M&M?

-Fig. 6. It is a bit confusing that an inverse topology is used to show KdpD (from the Ct to the Nt, left to right), as compared to the previous figures. The authors should harmonize that and inverse the topology of KdpFABC (using a mirror image) to put it to the left hand side of the KdpD scheme.

-Line 451 no Table S1 was provided

-Fig. S3 : KdpNTD could have been further purified (eg size exclusion chromatography because the AIEX does not seem to be very efficient).

-Legend to Supplementary Figure 7 : line 66 : " 'E', Purification of D". Should be replaced by 'f'

-Line 67 f, g, should be replaced by g,h

Jean-Michel Jault

REVIEWER COMMENTS

Reviewer #1 (Remarks to the Author):

With great interest I was reading the manuscript by Silberberg and coworkers about the control of potassium uptake in the Gram-negative model bacterium *Escherichia coli*. *E. coli* possesses several uptake systems potassium, which is the most abundant cellular cation in many organisms. In *E. coli* as well as in other organisms, the rapid uptake of potassium ions is needed for the survival of the cell when it encounters hyperosmotic growth conditions. The *E. coli* KdpFABC potassium uptake system has a high affinity for the osmolyte. Therefore, the activity of the system needs to be tightly regulated to prevent the uptake of potassium ions to toxic levels. Since a while it is known that the KdpB subunit of the KdpFABC system can be phosphorylated. The phosphorylation of KdpB is required to shut down potassium uptake. However, the kinase responsible for the inactivation of the KdpFABC system was so far unknown.

By employing a series of elegant genetic and biochemical approaches Silberberg and coworkers demonstrated that the sensor kinase KdpD of the KdpDE TCS (that also controls the expression of the *kdp* genes) can phosphorylate KdpB, thereby inactivating the KdpFABC system. Since a long-standing open question has been solved the work is a very important contribution to the field of osmoregulation.

We thank the Reviewer for their enthusiastic and insightful comments. We have implemented the suggested corrections and additions as follows:

I have only a few suggestions that should be addressed by the authors to improve the manuscript.

Major points

1. In my point of, the title of the manuscript is misleading. I would suggest to change the title as follows: "Sensory kinase KdpD is a tandem serine histidine kinase controlling K⁺ pump KdpFABC on the transcriptional and post-translational level". KdpD is required to activate the transcription of the *kdp* genes (transcriptional level regulation) and KdpD phosphorylates and inactivates the KdpFABC system (post-translational level regulation).

This is correct and has been adjusted accordingly.

2. Phylogenetic analysis of the Kdp system. Comparative genomics can often lead to wrong conclusions due to sequencing errors and wrong annotations. For instance, it has been wrongly described that in some organisms the *cdaAR* genes are fused to one gene encoding a longer protein that harbors a diadenylate cyclase (DAC) domain and the regulator of the DAC. Due to sequencing errors this assumption is wrong. Did the authors consider this problem when analysing the sequences of KdpD orthologs?

It is true that there are multiple cases where sequences in the Uniprot database appear to be incomplete. We nonetheless noted these down according to the domain structure in the sequences available, but these cases are comparatively rare (falling in the 'other' category in Figure 4 a, totalling less than 2% of sequences analyzed).

The methodology used is also dependent on the correct annotation of sequences as KdpD in the Uniprot database. However, wrongly assigned, non-KdpD hits were excluded based on sequence alignments. It remains likely that there are KdpD sequences that are not annotated in Uniprot, which could therefore not be included in the analysis.

To address these issues we have added the following sentences: "Notably, this dataset is dependent on annotations and sequences in the Uniprot database. While faulty annotations were removed by sequence alignments, incomplete sequences cannot be excluded. However, the frequency of the five most common classes of KdpD in the dataset suggests that a sequencing artefact is, in these cases, highly unlikely."

3. Discussion section: The tight regulation of the KdpFABC system at the transcriptional and post-translational level strongly resembles the situation in *Bacillus subtilis* and other related species. In *B. subtilis*, the KtrAB system is regulated at the transcriptional and post-translational level by the second messenger c-di-AMP. The authors might consider mentioning this in the discussion.

We thank the reviewer for this suggestion, and agree that placing the role of c-di-AMP discovered here into a larger context of c-di-AMP control of K⁺ homeostasis is an important point to make.

Therefore, we have added the following:

"The full control of some Kdp systems by c-di-AMP is reminiscent of the regulation of K⁺ transport systems KtrAB and KimA in *Bacillus subtilis*, whose expression is controlled by c-di-AMP binding to the *ydaO* riboswitch and which are additionally allosterically regulated by the second messenger^{44,50-54}. This works adds a new role for c-di-AMP in K⁺ homeostasis, highlighting the importance of the second messenger in bacterial adaptability."

Minor points

Page 1, line 29: The Usp domain is not explained in the abstract. Please either explain or delete.

The Usp domain has been deleted from the abstract.

Page 2, lines 47 and 50: better "kdpFABC".

The correction has been implemented.

Page 3, line 84: better "that controls the transcription of the kdpFABC genes and directly phosphorylates..."

The correction has been implemented.

Page 6, figure 2: The gel figure seems to be cut at the left site. Please explain!

The gel additionally includes a molecular weight marker, which is not shown in the cropped band in Figure 2. The full, uncropped Phosphoprotein and Coomassie gels are shown in Supplementary Figure 2.

Page 11, line 251: better "indicating that the Walker B motif..."

The correction has been implemented.

Page 13, line 308: "a second messenger, which..."

The correction has been implemented.

Page 13, line 314: Please provide a reference here.

A reference has been added to Gundlach et al., 2017 (doi: 10.1126/scisignal.aal3011), which shows that c-di-AMP is accumulated at high K⁺ compared to low K⁺ conditions.

Page 15, line 351: better "ASK"

The correction has been implemented.

Page 28, line 668: "B. Macek"

The correction has been implemented.

Reviewer #2 (Remarks to the Author):

K⁺ homeostasis plays a major role in the maintenance of bacterial osmolality. As a high-affinity K⁺ transport system, the expression of KdpFABC is tightly regulated by a two-component system, KdpDE. In this manuscript, the authors found that KdpD, a previously described HK of the TCS KdpDE, additionally mediates an inhibitory serine phosphorylation of KdpFABC at high K⁺ levels, using not its C-terminal HK domain but an atypical N-terminal serine kinase (ASK) domain. This finding reveals a new regulatory mechanism in the K⁺ transport system: at high K⁺ concentrations, KdpD not only repressed kdpFABC expression at the transcriptional level, but also irreversibly inhibited KdpFABC transport at the post-translational level by phosphorylating KdpB. They also analyzed and compared the structure of KdpD in bacteria and revealed different regulation of KdpD, e.g., in *E. coli*, the full-length KdpD is controlled by the concentration of K⁺, whereas in *Deinococcus geotherm*, the kdpD, which contains only USP and ASK domain, can be regulated by c-di-AMP, which affects the phosphorylation of KdpB. The finding adds to the diversity of sensor kinase functions and expand the framework of bacterial K⁺ regulatory mechanisms. I have some comments listed below.

We thank the reviewer for their helpful comments, which we have addressed as follows:

1. The rapid, irreversible inhibition of KdpFABC at high K⁺ is widely conserved and mediated directly by the KdpD's ASK domain. Since there is no covalent bond formation between KdpFABC and KdpD ASK domain, why the inhibition by ASK is irreversible?

The inhibition was previously shown to be irreversible *in vivo* (doi: 10.1046/j.1365-2958.2000.01793.x), as referenced in the introduction. Previous works have shown that dephosphorylation of the pump by lambda-phosphatase is possible *in vitro* (doi: 10.1038/nature22970), suggesting that it is only the lack of a specific phosphatase for KdpFABC in *E. coli* that makes the inhibition mechanism irreversible.

2. In Fig. 5, DDM or SMALPs were used to mimic the native membrane environment and for EckKdpD or DgKdpD activity assay. Yet, in Fig. 5b, when DDM was used to mimic membrane environment, the phosphorylation activity of EckKdpFABC seems to be attenuated in the presence of 400 mM K⁺, not simulated as described.

Previous works have shown that, *in vivo*, both the HK activity of KdpD and the phosphorylation of KdpB are K⁺ dependent (doi: 10.1074/jbc.272.16.10847 doi: 10.7554/eLife.55480), but when purified in DDM, the HK activity is decoupled from K⁺, suggesting that DDM is not a suitable membrane mimic. This observation is confirmed for the serine kinase activity by our results here.

To clarify, we have rephrased and added a Figure reference as follows: "In vivo, the phosphorylation of KdpB_{S162} in *E. coli* and the HK and phosphatase activities of KdpD are dependent on K⁺ levels^{12,14}. However, when purified in DDM, the HK activity is constantly low, while the phosphatase activity is high¹⁰. In agreement with this observation, KdpD solubilized in DDM here showed a constantly high serine kinase activity (Figure 5 B). Since a membrane environment was shown to restore the K⁺ dependence of KdpD's HK activity¹⁰, we solubilized KdpD directly from *E. coli* membranes in SMALPs to retain the K⁺ stimulation of the serine kinase activity."

3. Also, in Fig. 5, DDM was used to mimic the native membrane environment. Does DDM forms a micelle in this case? When EckKdpD is reconstituted into this micelle, how can one control its orientation? Is the KdpD/Usp domains exposed to water or buried inside the micelle? Can K⁺ penetrate the micelle to control the KdpD ASK activity if they are buried inside the micelle?

While we have not solved a structure of KdpD in detergent, we assume that it will behave much like other membrane proteins in detergent, i.e. DDM will form a belt around only the hydrophobic sections of the TM helices, while the soluble domains (KdpD, Usp, GAF, and DHpCA) and the solvent-protein interface remain solvent-accessible.

Since each protein has its own micelle that keeps it in solution, orientation is not applicable (there is no separate compartment formed by the micelle).

4. Also, in Fig. 5, the DgKdpD system is water soluble, thus no DDM or SMALPs is required to do the assay. But again, the K⁺ seem to substantially attenuate the phosphorylation activity of DgKdpD. Where does K⁺ bind to affect this activity?

We have observed K⁺ attenuation of the phosphorylation signal in multiple K⁺-insensitive systems (*E. coli* KdpD in DDM, *D. geothermalis* KdpD, and the *E. coli* NTD by itself (not shown)). Our assumption is that the high salt concentration used (400 mM) weakens interprotein interactions, reducing the kinase efficiency. This would be similar to e.g. a high-salt wash step during protein purification. Alternatively, the high salt content could influence the running behavior or phosphoprotein stain of the SDS gel. Currently, we have no evidence that this attenuation is a K⁺-specific effect.

5. In Fig. 6c, why does the addition of K⁺ make KdpB phosphorylation decrease? Are there other pathways by which K⁺ affect KdpB phosphorylation?

We believe you are referring to Figure 5 c, the effect of which is explained in the response to point 4.

Figure 6c outlines the regulation of KdpDs ASK activity by c-di-AMP at low K⁺. At low K⁺, c-di-AMP levels are low, so the kinase activity is not significantly stimulated. We have no evidence for other pathways by which K⁺ affects KdpB phosphorylation in these species.

6. The authors found that the N-terminal KdpD domain of KdpD is an atypical serine kinase (ASK) domain, but it is not labeled in any figure?

We have opted for the commonly used domain nomenclature for KdpD in figures, where the N-terminal domain is called KdpD domain. However, we like your suggestion, which helps clarify the new role identified for this domain. Therefore, after our classification of this domain as an ASK, we have now added an ASK annotation to Figures 4 and 5.

7. Please describe in detail how DgKdpD is regulated by c-di-AMP?

Our current knowledge on the regulation of *DgKdpD* by c-di-AMP is summarized in paragraph 4 of the discussion. The exact molecular mechanisms by which c-di-AMP binding to the Usp domain activates the ASK activity are unclear and require significantly more structural and functional data, which for us has been challenging so far, and which would therefore exceed the scope of this manuscript. We appreciate the reviewer's enthusiasm for this question, and hope to be able to answer it in the future.

8. An evolutionary tree should be drawn to elucidate the functional differentiation of KdpD protein.

We agree with the reviewer that the evolutionary links between the different classes of KdpD identified here are fascinating. However, the potential evolutionary links between KdpD versions outlined in the discussion are purely speculative – we feel that an evolutionary tree as a figure could easily be misconstrued as fact, for which we do not have enough data.

9. Typo: line 92: controlled either by directly by the K⁺ concentration or by

The correction has been implemented.

10. Supplementary Fig. 6c (line 316) is not visible.

The incorrect reference was updated to "Figure 4 D"

11. Please briefly explain DDM and SMALP in the text.

To clarify the difference between DDM and SMALPs, we have added the following sentence: "Contrary to detergents like DDM, which displace lipids to form micelles tightly around the TM domain, SMALPs can form stable lipid discs containing the membrane protein."

Reviewer #3 (Remarks to the Author):

This manuscript by Silberberg et al. describes a very interesting dual activity borne by a His-Kinase, KdpD, previously known to be part of a two-component systems which controls the expression of a K⁺ pump, KdpFABC. Here, it is shown that in addition to the classical His-kinase function, KdpD is also capable to phosphorylate the Ser162 residue of the KdpB subunit of the K⁺ pump by an atypical serine kinase (ASK) to regulate post-transcriptionally the function of the pump. This ASK activity is borne on a N-terminal domain of the KdpD and in some bacterial species, this domain is only fused to a Usp domain, no HK domain is therefore present. In such a case, instead of being controlled by the K⁺ level, as in full-length KdpD, the ASK can be controlled by a second messenger such as cyclic di-AMP, as shown here from the *Deinococcus geothermalis* enzyme. This result clearly expands the knowledge on how the function of a K⁺ pump is optimally controlled both at the translational and post-transcriptional level by a dual kinase, sometimes borne on different polypeptide, and that this regulation is finely tuned by the use of different intra- or extra-cellular signals.

Overall, the experiments showing the new ASK activity are well-performed and quite convincing and the results reported clearly warrant to be published.

We thank the reviewer for their kind comments and helpful insights.

However, I have a major concern regarding the lack of ATPase activity reported for the ASK domain. The amount of protein used (0.5-1 microgr.) is really too low if one wants to detect a low level of ATPase activity. It has been reported that some protein kinases do exhibit a very low level of ATPase activity when no protein substrate is provided in vitro (e.g. 10h⁻¹, please see Nguyen et al. *J Mol Biol.* 2017 Oct 13;429(20):3056-3074. doi: 10.1016/j.jmb.2017.08.016.). This does make sense as some water molecules near the catalytic site can promote this low level of ATPase activity in the absence of a protein substrate. So before ruling out any ATPase activity of the ASK domain, the authors need to perform the ATPase assay using much more enzyme (e.g. 20 ug) and with a longer incubation time (e.g. 30 minutes at 37°C).

We have performed the ATPase assay with 15 µg KdpD_{NTD}, 12 mM ATP, and for 30 min at 37 °C, which yielded an ATPase rate of around 2 µmol P_i g⁻¹min⁻¹ (see below). This extremely low level of residual ATP hydrolysis (per gram!) indicates that ATPase activity is most likely mechanistically irrelevant under physiological conditions. The remainder of ATPase activity was lost when non-phosphorylatable substrate was added at equimolar levels, indicating that, when water is excluded, no free phosphate is generated at all. Thus, a mechanism with a free phosphate intermediate can be excluded. Supplementary Figure 5b has been exchanged and the figure caption adapted accordingly.

Also, many Ser/Thr/Tyr kinases, even aspecific ones, are capable to use classical 'in vitro' kinase substrates such as the myelin basic protein. So it would be nice that the authors try to phosphorylate such a kinase substrate in vitro using full-length KdpD and the catalytic ASK domain. This will nicely corroborate the Ser kinase activity of KdpD.

We appreciate this suggestion. However, we do not see the advantage of this experiment. The experiments performed already show that KdpD phosphorylates KdpB_{S162}, including a selectivity over other sidechains (KdpB_{S162T/S162Y}). Therefore, we hope that the reviewer agrees that the assignment of the KdpD domain as a serine kinase is clear, and phosphorylation of another (physiologically not relevant) substrate does not contribute much to the conclusions drawn.

Minor remarks:

-Line 92 : "by directly by the": remove the first 'by'.

The correction has been implemented.

-Fig. 2. Please show the membrane limit on the scheme, color code the names of the subunit according to the figure, and use different symbol for the D307 and S162. The activity is normalized to 100% but indicate the specific activity in the legend

We thank the reviewer for the suggested improvements have implemented them as proposed. The normalization of the activity has been clarified in the legend.

-Why do you need to use a triple mutant KdpDG36A/K37A/T38C in the Walker A motif, referred to as KdpDΔWalkerA? Usually, a point mutation of the Lysine of the WA is quite enough to kill the activity?

We used this triple mutant of KdpD because it was previously used in experiments investigating ATP binding to the NTD (doi 10.1074/jbc.273.28.17406). Based on the similarity of point mutations in the Walker B motif to other Walker A/B proteins, it is likely that a mutation of only K37 would be sufficient to abolish the kinase activity.

-Please, show as Supplemental data an alignment of KdpD highlighting the Walker A and B motifs and all the other residues mutated afterwards.

We thank the reviewer for this helpful suggestion. We have added a new Supplementary Figure 5 showing the conservation of the Walker A/B motifs, the nucleotide-binding tryptophan-arginine pair, and the catalytic histidine of the HK.

-Line 158 : Ref 32 is written in parenthesis.

We formatted references following a superscript (e.g. K⁺) differently to ensure that they are recognizable. This will be standardized according to the journal's guidelines during final formatting of the manuscript.

- Line 268: the ASK and HK activities can be borne on separate polypeptide chains: are they part of the same operon in this case?

We thank the reviewer for this insightful question. Upon closer examination, we have found that the ASK-encoding gene is featured downstream of the pump, while the HK is separate. This means that the ASK is also highly expressed when the pump is produced by read-through expression, providing the required amount of ASK-KdpD for a rapid inactivation of KdpFABC when the time comes. We have added the following in the discussion to reflect these insights:

"In some species such as *Deinococcus radiodurans*, *Anabaena sp.*, and *Alicyclobacillus acidocaldarius*, *kdp* expression is regulated by a separate TCS⁵⁵⁻⁵⁷. Interestingly, in these species, the ASK-containing chain is retained downstream of the pump in the *kdp* operon, while the HK is separate. This indicates that the read-through expression of the ASK is important to ensure sufficient protein levels to efficiently inhibit the pump when required."

- Fig 4. What the group 'others' stands for ? Are the sequences truly related to KdpD?

We thank the reviewer for highlighting this point, which we agree profits from further clarification. 'Other' represents all sequences that do not fall into the most common five classes of KdpD found. Many of these sequences appear to be the result of incomplete sequencing, which is a caveat to identifying KdpD sequences through Uniprot. In total, species in which these different fragments are annotated add up to around 2%.

To address these issues we have added the following sentences: "Notably, this dataset is dependent on annotations and sequences in the Uniprot database. While faulty annotations were removed by sequence alignments, incomplete sequences cannot be excluded. However, the frequency of the five most common classes of KdpD in the dataset suggests that a sequencing artefact is, in these cases, highly unlikely."

- Line 316 : There is no Supplementary Fig. 6c provided with the MS.

The incorrect reference was updated to "Figure 4 D"

-Fig. 5: is it 5 mM c-di-AMP or 1 mM as mentioned in Supplemental data and in the M&M?

Experiments were done with 1 mM c-di-AMP, which has been corrected in the legend of Figure 5.

-Fig. 6. It is a bit confusing that an inverse topology is used to show KdpD (from the Ct to the Nt, left to right), as compared to the previous figures. The authors should harmonize that and inverse the topology of KdpFABC (using a mirror image) to put it to the left hand side of the KdpD scheme.

Thank you for this helpful comment. We agree and have modified the figure as suggested.

-Line 451 no Table S1 was provided

The reference was corrected to "Table 1"

-Fig. S3 : KdpNTD could have been further purified (eg size exclusion chromatography because the AIEC does not seem to be very efficient).

While in this case the purity is not ideal, the identity of KdpD as the direct kinase for KdpB had already been established. Therefore, the purity is sufficient for the subsequent functional assays shown.

-Legend to Supplementary Figure 7 : line 66 : " 'E', Purification of D". Should be replaced by 'f'

The correction has been implemented.

-Line 67 f, g, should be replaced by g,h

The correction has been implemented.

Reviewer #1 (Remarks to the Author):

I am happy with the revised version of the manuscript NCOMMS-23-62418A. I fully support the publication of the manuscript by Silberberg and co-workers in Nature Communications.

Fabian Commichau

Reviewer #2 (Remarks to the Author):

The quality of the manuscript has improved, and I have no further request for revisions.

Reviewer #3 (Remarks to the Author):

The authors have taken into account most of my remarks.
This very nice manuscript can now be published.

Jean-Michel Jault